# Does a Gluten-Free Diet Affect BMI and Glycosylated Hemoglobin in Children and Adolescents with Type 1 Diabetes and Asymptomatic Celiac Disease? A Meta-Analysis and Systematic Review

**DOI:** 10.3390/children9081247

**Published:** 2022-08-18

**Authors:** Salma Burayzat, Nour Elsahoryi, Ali Freitekh, Osama Alzoubi, Rahaf Al-Najjar, Reema Tayyem

**Affiliations:** 1Pediatric Gastroenterology, Hepatology and Nutrition, Faculty of Medicine, Pediatrics Department, The Hashemite University, Zarqa 13133, Jordan; 2Faculty of Pharmacy and Medical Sciences, Nutrition Department, University of Petra, Amman 961343, Jordan; 3Faculty of Medicine, The Hashemite University, Zarqa 13133, Jordan; 4Faculty of Medicine, Jordan University, Amman 11942, Jordan; 5Department of Human Nutrition, College of Health Sciences, Qatar University, Doha P.O. Box 2713, Qatar; 6Faculty of Agriculture, University of Jordan, Amman 11942, Jordan

**Keywords:** type 1 diabetes mellitus, celiac disease, gluten-free diet, body mass index, HA1C

## Abstract

**Background:** Children diagnosed with type 1 diabetes mellitus (T1DM) are more prone to having celiac disease (CD) than the normal population. Moreover, patients with this dual diagnosis who are also on a diabetic and gluten-free diet (GFD) risk faltering growth and uncontrolled blood glucose levels. This review aims to assess the efficacy and effectiveness of managing patients with T1DM screened for CD with GFD to prevent complications associated with these chronic pathologies in childhood and adulthood. **Materials and Methods**: We abided by the PRISMA guidelines in this meta-analysis and used multiple databases and search engines. We included case–control studies. The primary outcomes were changes in the standard deviation score, body mass index (SDS BMI), and glycosylated hemoglobin (HA1C) after being on a GFD for at least twelve months. **Results:** The pooled data from the six studies included showed that there was neither a statistically significant difference in the mean SDS BMI (−0.28 (95% CI −0.75, 0.42)) (*p* = 0.24) nor in the mean of HA1C (mean −0.07 (95% CI −0.44, 0.30)) (*p* = 0.36) for the same group. HDL cholesterol improved significantly in patients on a strict GFD (*p* < 0.01). **Conclusions:** In children with T1DM and asymptomatic CD, being on a GFD had no significant effect on BMI or HA1C. However, it can have a protective effect on the other complications found in both chronic pathologies.

## 1. Introduction

The European Society for Pediatric Gastroenterology, Hepatology and Nutrition (ESPGHAN) [1] defines celiac disease (CD) as a chronic immune-mediated systemic disorder driven by gluten consumption from wheat, rye, and barley and their derivatives, which might also contaminate other grains, in the presence of specific genetic susceptibility. It presents a variable combination of clinical symptoms, specific serological markers, an HLA-DQ2/DQ8 haplotype, and enteropathy. Life-long strict adherence to a gluten-free diet leads to the disappearance of symptoms, negative titers of autoantibodies, histological recovery of the intestinal mucosa, and the elimination of long-term sequelae [2].

CD is diagnosed in pediatric patients diagnosed with type 1 diabetes mellitus (T1DM) more frequently than in the general population. In T1DM patients, CD prevalence can be up to twenty times higher than in the general population. The prevalence of CD in patients with T1DM is 3–12% [3,4].

Despite this increased risk, many health care providers struggle to reach an optimal approach for managing CD in T1DM to prevent the short- and long-term complications of these two life-long pathologies, especially in the absence of a consensus to guide their management suggestions [1]. 

Diagnosing patients with symptomatic CD, including malabsorption, and patients with subclinical disease, detected via osteopenia, growth failure, hepatic dysfunction, menstrual irregularity, unexplained epilepsy, or ataxia, remains uncomplicated. A gluten-free diet (GFD) in these symptomatic children improves their presenting signs and symptoms, making this regime strongly recommended. 

Serological autoimmune markers such as the IgA anti-tissue transglutaminase (TTG) and IgA anti-endomysial (EMA) antibodies are highly sensitive and specific, and they are now used for routine CD screening to identify ‘silent’ and ‘atypical’ forms of CD. Many of the patients identified by screening are usually asymptomatic. Evidence is inconclusive concerning the advantages versus the disadvantages of screening and treating asymptomatic individuals who are children that are already burdened with an established chronic illness [5,6].

As the development of T1DM usually precedes symptomatic and asymptomatic CD, systematic CD screening should be performed periodically in all children with T1DM [3].

GFD has been proven to benefit growth and nutrient absorption in patients with CD. On the other hand, it is unclear whether the institution of GFD in T1DM and subclinical CD patients is beneficial. Despite many screening studies on this subject, only a few have tackled the effects of putting T1DM and subclinical CD patients on a GFD. Only a handful of small prospective and retrospective studies have addressed the glycemic benefits of a GFD. Given the limited information on the role of GFD in subclinical CD cases, we decided to perform a meta-analysis and systematic review of observational case–control studies conducted on children with T1DM and asymptomatic biopsy-proven CD [7]. Children were asked to follow a GFD for at least twelve months. The primary objectives were to document the pooled effects, regardless of whether they were positive or negative, of a GFD on anthropometric parameters, mainly body mass index (BMI) and glycosylated hemoglobin (HbA1c) levels, for this group of patients. Secondary objectives were to report any other effects of a GFD in these patients, such as albumin excretion in urine, insulin dose requirement, and bone mineral homeostasis. 

To date, no review on the benefits of a GFD in children with a dual diagnosis of T1DM and asymptomatic CD has been reported before this one. We used Grading of Recommendations Assessment, Development, and Evaluation (GRADE) methods to assess the overall quality of evidence, and this method was not used in past reviews, Table A1. This systematic review and meta-analysis were registered in PROSPERO.

## 2. Materials and Methods

The studies included in this review were prospective and retrospective case–control studies; randomized control studies could not be found, as confirmed cases of CD should always be managed by a GFD and cannot be randomized to placebo management. This meta-analysis is reported according to the Preferred Reporting Items for the Systematic Reviews and Meta-Analyses (PRISMA) [8] statement and checklist (Appendix A).

The study protocol was pre-specified and registered in PROSPERO (ID: CRD42020186088) and submitted on the 5 July 2020. The registration record was published exactly as submitted. The PROSPERO team did not check for eligibility based on an exclusive focus on COVID-19 registrations during the 2020 pandemic.

We applied population, intervention, comparison, and outcome (PICO) characterization to operationalize the research questions and to determine the eligibility criteria. The inclusion criteria were as follows: children aged between one and eighteen years, an established diagnosis of T1DM at least one year before the diagnosis of CD, CD diagnosed and proven by specific antibodies and small intestinal biopsy, and prescription of a GFD for at least one year before the last assessment and follow up. As the control group of each study did not have CD, they were not put on a GFD. Studies with multiple outcomes were included. In our study, two different primary results obtained before and after adherence to a GFD for at least twelve months were studied in this review: body mass index SDS (BMI SDS) [9] was studied as an anthropometric index, and glycosylated hemoglobin (HA1C) was studied as a metabolic index. BMI SDS was used as an anthropometric index, as growth influences body weight and adiposity measurements during childhood; it is more easily standardized into SD scores (SDS) with respect to reference populations. Standard glycemic parameters such as blood glucose and glycosylated hemoglobin are used to monitor control in diabetic patients. New methods, such as glycemic variability, are emerging to aid in better management and monitoring. Most of the research in this domain still uses glycosylated hemoglobin as a metabolic parameter to follow in diabetics [10].

This review’s secondary outcomes were renal function, lipid profile, bone density, and hypoglycemic events.

The authors confirmed adherence to the GFD in subjects in the selected studies for the meta-analysis by considering the lack of symptoms in the studied subgroups. T1DM was diagnosed, and insulin therapy was initiated at least one year before confirming CD diagnosis to eliminate the effects of insulin therapy initiation on the weight and BMI of the patients. These two inclusion criteria were limiting factors in finding eligible studies.

Studies were identified using electronic sources. The following databases were used for the electronic searches: PubMed, EBSCO, the Cochrane Library, the Cochrane Central Register for Controlled Trials, Google scholar, and Springer (Appendix A). The investigation was conducted between the following dates: 28 September 2019 and 31 January 2020. Literature search strategies were developed using medical subject headings (MeSH) and free text. The complete electronic search strategy for all of the databases is presented in detail in the Appendix A.

No restrictions based on outcomes or publication were applied to the searches. The only rules applied to the searches were studies on humans, language (English and French studies included), and age from one to eighteen years. 

Three authors independently screened the titles/abstracts of potential studies for inclusion of the pre-specified inclusion/exclusion criteria. Following the initial screening, the studies’ full texts were reviewed using the same inclusion/exclusion criteria as the initial screening. A third reviewer was consulted to evaluate a study’s inclusion whenever there was a conflict between the two authors.

Studies meeting the following inclusion criteria were included in the study: study population of pediatric age group, a control group with a diagnosis of T1DM, CD confirmed by small intestinal biopsy, and patients followed for at least one year.

We assessed the quality of the overall evidence using the GRADE approach. This quality assessment method considers the study type, within-study risk of bias (methodological quality), heterogeneity, directness of evidence, precision of effect estimates, and publication bias risk. We rated the quality of the body of evidence for each key outcome as ‘high, moderate, low,’ or ‘very low’, Table A1 [11].

## 3. Statistical Analysis

The mean and standard deviation scores (SDSs) of the included studies were then input into Cochrane Review Manager Software, RevMan Version 5.4 (The Cochrane Collaboration, Copenhagen, Denmark). Weighted mean differences between the T1DM with CD groups at diagnosis and follow-up after being put on a GFD were calculated for BMI and HbA1c with 95% CIs using the random effects. In the random-effects analysis, we presumed that the actual effect size varied from one study to another. The studies in our analysis represent a random sample of effect sizes that could have been observed. The summary effect estimates the mean of these effects and accounts for heterogeneity in participant populations. Tests to determine slope heterogeneity between studies were performed, and forest plots were assessed using the x2 test and the I^2^ statistic. I^2^ values below 25% were considered to have no heterogeneity, and values up to 40% were not considered to represent significant heterogeneity. Funnel plots were performed to graphically assess potential publication bias, which was statistically evaluated with Egger’s test. Moreover, a one-way t-test was used to calculate the significant difference in the standard deviation scores (SDSs) for BMI and HbA1c among both arms (CD and T1DM at diagnosis and follow-up). A *p*-value of <0.05 was considered statistically significant for testing the pooled results of the included studies and for univariate analysis. 

## 4. Results

### 4.1. Characteristics of Included Studies

Our literature search yielded 552 titles, as shown in Figure 1. According to the PRISMA guidelines, studies were removed due to duplication, not meeting the inclusion criteria, or the statistical variables not being unified. We included six case–control studies [12,13,14,15,16,17] in the meta-analysis, as summarized in Table 1. There were five prospective [12,14,15,16,17] and one retrospective [13] case–control studies on the effects of a GFD on patients with T1DM and CD involving 578 participants. In the six studies, children who had T1DM and were found to have high titers of screening antibodies for CD and who underwent small bowel biopsies were put on a GFD and matched to children diagnosed with T1DM according to age and gender. The control groups were not put on a GFD, as the children in these groups did not have CD.

Three studies were carried out in the United Kingdom [12,13,15]. The other three were carried out in Europe [14] (in 10 pediatric diabetic centers around Europe), Australia [16], and Italy [17]. 

The primary outcome results, including the basic demographics and inclusion/exclusion criteria from the six studies, were reported in peer-reviewed journals [12,13,14,15,16,17]. The number of patients varied between studies, as shown in Table 1. The study of Rami et al. [14] had the highest number of participants (n—293 at baseline and 269 at follow-up), while the study of Amin et al. [15] had the lowest number of participants (n—33) at baseline and at follow-up. However, all of the included studies had no significant loss to follow-up, except Rami et al. [14], which had 0.1% of the participants withdraw. 

The patients’ age at CD diagnosis varied among the studies, ranging from 7.5 years in Saadah et al.’s study [16] to 13.8 years in Amin et al.’s study [15]. Regarding gender, the same percentage of females was distributed among the two groups, except in Rami et al. [14], where females accounted for 45% of the cases and 50% of the comparators. The follow-up time varied among the studies, but all of the studies had at least one year of follow-up, with Amin et al. having a longer follow-up period of 4 years [15]. According to GRADE guidelines, three studies [11,12,13] had a low risk of bias, and the other three [8,9,10] had a moderate risk of bias, Table A1. 

Concerning the age of T1DM diagnosis in the group with a dual diagnosis of T1DM and CD, only one study, Saadah et al. [16], showed an earlier T1DM diagnosis in the groups of T1DM and CD patients in comparison to the control group with T1DM. The age range at T1DM diagnosis was between four and eight years, while the range of the diagnosis age for CD was between three and seventeen years. The average duration of T1DM diagnosis before CD diagnosis was 2.3 years (1–3.1 years). 

### 4.2. Body Mass Index—SDS 

Meta-analytical findings of the first condition are described in the forest plot, as shown in Figure 2. The pooled results of the overall effect of the six studies included in this research indicated that there was no statistically significant difference in the mean (0.00 (95% CI −0.03, 0.02)) of the SDS BMI of the T1DM patients between baseline and follow-up (*p* = 0.22). The pooled results indicated no significant heterogeneity between studies (*p*-value = 0.47). The index to quantify the dispersion of the effect (I^2^) statistic = zero, reflecting minimal heterogeneity. For the T1DM and CD patients, the pooled results of the overall effect studies indicated that there was no statistically significant difference in the mean (−0.28 (95% CI −0.75, 0.42)) of the SDS BMI for the T1DM group with CD between baseline and follow-up (*p* = 0.24). The pooled results indicate that there was significant heterogeneity between studies (*p* ≤ 0.001), and the index to quantify the dispersion of the effect (I^2^) statistic = 98%, which reflects significant amounts of heterogeneity, as shown in Figure 3. 

### 4.3. Glycosylated Hemoglobin

Regarding HbA1c, as shown in the forest plot in Figure 3A, the pooled results of the overall effect of the six studies indicated that there was no statistically significant difference in the mean (mean difference −0.08 (95% −0.23, 0.07)) of the SDS BMI of T’DM patients between baseline and follow-up (*p* = 0.72). The pooled results indicate significant heterogeneity between studies (*p* ≤ 0.001), and the index to quantify the dispersion of effect (I^2^) statistic = 96%, which reflects significant heterogeneity. As shown in Figure 3B, the overall impact of both groups indicated that there was no statistically significant difference in the mean (mean difference −0.07 (95% CI −0.44, 0.30)) between baseline and follow-up ((*p* = 0.36). The pooled results indicate significant heterogeneity between studies (*p* ≤ 0.001), and the index to quantify the dispersion of effect (I^2^) statistic = 96%, which reflects significant heterogeneity. 

### 4.4. Lipid Profile, Hemoglobin Levels, Diabetic Renopathy, and Retinopathy 

As a secondary outcome, an improved lipid profile (HDL, total cholesterol, and triglyceride) after strict adherence to a GFD for at least six months was seen in three studies [18,19,20]. Hemoglobin and serum iron improved significantly after being on a GFD for at least six months [21,22]. Adherence to a GFD in patients with T1DM and CD might have a protective effect on the development of diabetic reno- and retinopathy [23,24,25,26], as seen in Table 2. 

## 5. Discussion

CD and T1DM are autoimmune pathologies where the body’s immune system produces autoantibodies that destroy its organs and tissues. These types of pathologies can affect any system in the body. Most of the time, more than one autoimmune disease can be present in the same patient concurrently, suggesting the necessity of a comprehensive management technique for multiple pathologies simultaneously. In this systematic review, a GFD did not significantly affect BMI-SDS and HA1c but showed the benefits of this specific diet on other health aspects, such as the lipid profile, diabetic retinopathy, and nephropathy.

CD and T1DM are highly associated with the HLA system, where they share common characteristics, such as the haplotypes A1, B8, DR3, and DQ2. The DQ2 locus, and mainly DQA1*0501/DQB1*, is found in over 90% of CD patients [40]. In patients with a dual diagnosis of CD and T1DM, the classic form of CD with typical gastrointestinal symptoms such as chronic diarrhea, body mass deficiency, abdominal pain, and flatulence may be observed in less than 25% of T1DM patients. However, extra-intestinal manifestations (short stature, iron deficiency anemia, or delayed puberty) and silent forms are more frequent. Recently published studies have shown that up to 71.4% of children with T1DM do not have gastrointestinal symptoms when CD-specific antibodies are detected [41,42]. International guidelines recommend CD screening at T1DM diagnosis and annually for five years [27]. In contrast, the Canadian Diabetes Association [28] recommends serologic testing based solely on clinical symptoms, including recurring gastrointestinal or extra-gastrointestinal symptoms and unexplained frequent low blood sugar levels. For adults, the recommendations are less specific [29].

In children, the most vital management technique for T1DM is dietary intervention, which can be challenging to follow and abide by in this age group [43]. In this particular group of patients, CD has a higher prevalence than in the normal population, and it is also managed by the annulment of gluten from the patient’s diet [44]. Combining these two strict dietary regimens (diabetic diet and GFD) in children can be very demanding, especially in the subgroup of patients with asymptomatic CD discovered by screening with specific antibodies and proven by small intestinal biopsy [30].

This meta-analysis and systematic review aimed to find an evidence-based approach to the effects of dietary management on these patients. The patients were diagnosed with T1DM for at least one year before CD diagnosis and starting a GFD to eliminate the effects of controlling blood sugar on weight at T1DM diagnosis that are associated with insulin therapy initiation. 

The collected data show that children with T1DM and CD lost weight after being on a gluten-free diet for at least one year. This weight loss could be explained by the difficulty of following two restrictive diets, but these studies failed to report a significant change in BMI SDS for these patients despite ensuring adherence to GFD [7,20,31]. 

Saadeh et al. illustrated that all of the anthropometric parameters of their populations were above the mean for the reference population, but none of them reached statistical significance. On the other hand, Acerini and colleagues [40] concluded that the positive benefits of dietary therapy in CD patients without gastrointestinal symptoms were uncertain after studying seven children with T1DM and CD being managed with a GFD. Westman and colleagues [45] found that dietary compliance did not change growth parameters in twenty children with T1DM and CD. Height was another anthropometric parameter that was studied. It should be noted that there were no significant changes observed in any of the studies involved in this systematic review [13,14,15,16,17,40].

This meta-analysis did not show any changes in the HA1c levels of children with a dual diagnosis of T1DM and CD after being put on a GFD. In the years before CD diagnosis, HA1c levels were usually lower than the levels measured after the confirmed diagnosis in the same patients [12]; this could be attributed to malabsorption in the absence of serum antibodies. Studies show that the presence of antibodies in the duodenal mucosa before their presence in the serum could be responsible for some of the elements of malabsorption [46,47]. However, some studies have shown an increase in HA1C levels at the beginning of GFD, but this might be explained by restoring the small intestinal mucosa and increased absorption capabilities. Nevertheless, this did not prove significant after a follow-up of at least one year [40,45,48,49].

Rami et al. [14] further subdivided the group of patients with dual diagnoses of T1DM and CD according to adherence to a GFD. Comparing the adherent and non-adherent to GFD groups, a trend toward a lower BMI SDS but not a higher z-score was noted in the non-adherent group, raising the notion of a positive influence of a GFD on weight gain in patients who adhere to a GFD.

Kaur et al. [32] reported a significant decline in the mean HbA1c level of 0.73% in the GFD group; on the other hand, it increased by 0.99% in the non-GFD group at the end of the follow-up period. Additional prospective studies have reported similar findings [15,48]. Meanwhile, other studies did not find any significant improvement in the HbA1c levels after GFD initiation [16,33]. 

Kaur et al. [32] did not report any differences in the insulin dose. At the same time, Packer et al. showed improvement in the HbA1c levels after GFD initiation compared to the controls despite no changes in the daily insulin dose being noted in their longitudinal prospective study. GFD might positively affect insulin sensitivity, explaining the improvement in HA1c values. Although the glycemic index of gluten-free and gluten-containing foods is similar [34], data suggest that the carbohydrate type might influence insulin sensitivity [50]. Moreover, for hypoglycemic events experienced before the diagnosis of CD, one study, the one by Rami et al. [14], reported no differences in the frequency of severe hypoglycemic episodes (loss of consciousness or abnormal movements) in both of the studied groups. The required insulin dosage also was not significantly altered in the group with bothT1DM and CD.

During this search, some studies did not fully meet the inclusion/exclusion criteria of this meta-analysis but still provided a growing body of evidence on the beneficial effects of a GFD for patients with concomitant T1DM and CD and indicated that it may protect against the development of other T1DM-related complications, as seen Table 2. 

Compared to T1DM patients without CD, Bakker et al. [26] showed that adult patients with TIDM and CD had a lower prevalence of retinopathy and lower total cholesterol than T1DM patients without CD. Warncke et al. [18] reported lower absolute systolic blood pressure in T1DM patients with CD than in those without CD. In the same cohort, patients with a dual diagnosis had significantly lower HDL cholesterol levels than individuals with T1DM alone at diagnosis; after the institution of a GFD, these levels increased significantly. The same effect was also noticed on bone density [51] as well as on hemoglobin and calcium levels [22]. A possible explanation for these improvements in cholesterol levels could be the normalization of the intestinal mucosa with the adoption of a GFD, demonstrating a beneficial effect of adhering to the diet.

Throughout our review, multiple papers have illustrated the possible reno-protective value of GFD on the progression of diabetic nephropathy. Malalasekera et al. [25] found that T1DM and CD patients had urinary albumin to creatinine ratios that were two-fold lower than those of patients with T1DM after adhering to a GFD for at least one year. Gopee et al. [24] reached similar findings of significantly lower urinary albumin to creatinine ratios in the same type of patients.

Although the pathophysiology of diabetic nephropathy is multifactorial, local inflammatory stress may result from both metabolic and hemodynamic derangements at a very early stage. Elmarakby et al. found significantly lower IL-1B, IL-4, and IL-5 levels in T1DM+CD patients who were adherent to a GFD than those with T1DM alone, suggesting a protective effect of a GFD on diabetic nephropathy progression [35,52]. 

Hypoglycemic episodes or a reduction in insulin requirements can be the presenting sign of CD in children with controlled T1DM. In contrast, a lack of hypoglycemic episodes at the clinical onset of CD could be the result of a higher mean blood glucose level [36]. In the six studies in this systematic review, only Rame et al. studied hypoglycemic episodes and insulin requirements. They found that no significant differences were documented after strict adherence to a GFD for at least one year of follow-up [14]. On the other hand, Mohn et al. reported a reduction in hypoglycemic events in children under a GFD [37]. 

Most of the studies in this systematic review, especially those with a large patient population, showed that a GFD normalizes the bowel mucosa and frequently leads to the disappearance of antibodies but may not necessarily lead to improved glycemic control, as seen in Table 2. 

Some studies showed a positive effect on growth parameters after adhering to a GFD for at least one year [18,19,21,38,39], yet others report no significant effect on the anthropometric indices [7,31,37,45,48,49,53]. On the other hand, long-standing CD may be associated with an increased risk of retinopathy [24], while non-adherence to a GFD may increase microalbuminuria risk [21,23,25].

A recently published study [54] reported that no significant adverse outcomes were found in children with T1DM with positive celiac serology who delayed therapy with a GFD for two years. On the other hand, the ISPAD states that in asymptomatic children with proven CD, a GFD can be considered justified to reduce the long-term risk of gastrointestinal malignancy and conditions associated with subclinical malabsorption (i.e., osteoporosis and iron deficiency) [55].

Although there were more than 500 studies found during our search due to the different search methods implemented to cover as much of the online databases as possible, only six studies met our strict inclusion criteria to decrease confounders and to have more robust results. Studies not included in online databases were not explored, as this study was not funded. Three of the six studies included in the study had low-quality evidence, and adherence to a GFD was inconsistently documented.

## 6. Conclusions

This meta-analysis and systematic review could not establish any significant positive or negative effects of a GFD on anthropometric indices or glycemic control. Meanwhile, none of the studies could individually prove a significant negative effect of a GFD on these patients. On the other hand, many clinical studies demonstrated the positive impact of a strict GFD in these patients on different health aspects, such as the possible benefit of a GFD being reno-protective and establishing a favorable atherogenic profile. Screening for CD in patients with T1DM is recommended, even in asymptomatic children. Untreated CD can lead to long-term sequelae in children with CD and in those with dual diagnoses of T1DM and CD. Further studies with large cohorts and control trials are needed to establish substantial evidence of the positive effect of GFD in children with T1DM and CD.

## Figures and Tables

**Figure 1 children-09-01247-f001:**
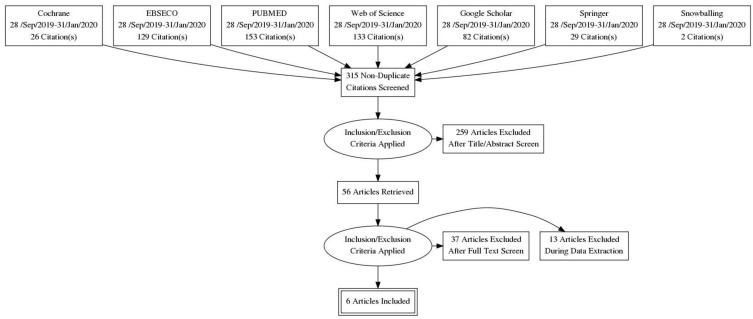
PRISMA flow diagram.

**Figure 2 children-09-01247-f002:**
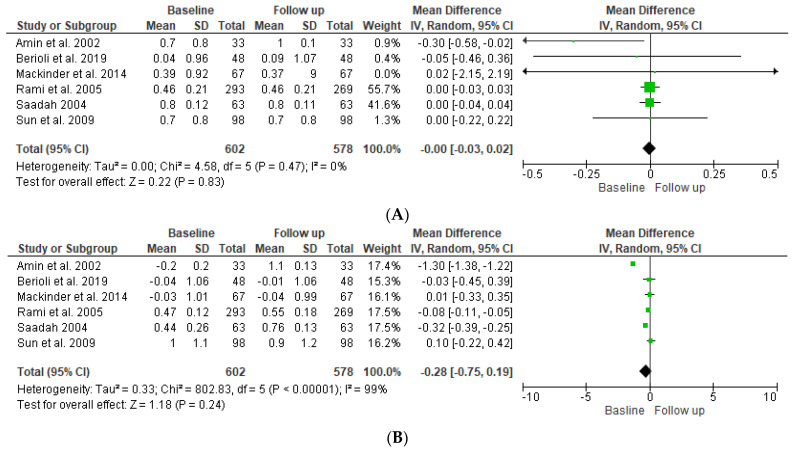
Forest plot showing standard deviation scores (SDSs) of the BMI for DM patients (**A**) and DM and CD patients (**B**) at baseline compared to follow-up [12,13,14,15,16,17].

**Figure 3 children-09-01247-f003:**
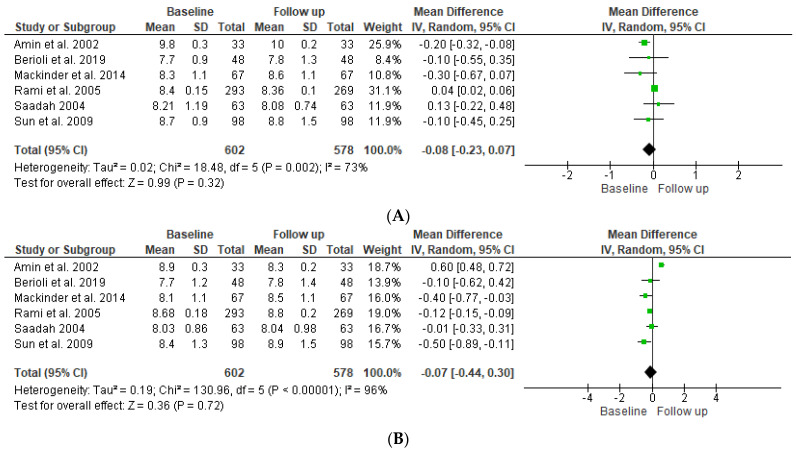
Forest plot showing HbA1c for T1DM patients (**A**) and T1DM and CD patients (**B**) at baseline compared to follow-up [12,13,14,15,16,17].

**Table 1 children-09-01247-t001:** Main characteristics of the studies included in the meta-analysis of the effects of GFD on BMI and HA1c in children with T1DM and CD.

	Sun et al., 2009 [12]	Mackinder et al., 2014 [13]	Rami et al., 2005 [14]	Amin et al., 2002 [15]	Saadah et al., 2004 [16]	Berioli et al., 2019 [17]
Country	United Kingdom	United Kingdom	Europe	United Kingdom	Australia	Italy
Study design	Prospective case–control	Retrospective case–control	Prospective case–control	Prospective case–control	Prospective case–control	Prospective case–control
Year	2009	2014	2005	2002	2004	2019
Number of participants/controls	49/49	18/9	98/195	22/11	21/42	16/32
Recruitment of cases and controls	Pediatric diabetic units in North West England	Outpatient clinic at Royal Hospital for Sick Children Glasgow	Ten pediatric diabetic centers around Europe	Pediatric Diabetic clinic at John Radcliffe Hospital, Oxford	Royal Children’s Hospital Melbourne	Regional Centre for Children with T1DM at the Pediatric Clinic of Universita Degli Studi di, Perugia
Female (%) in the cases/comparators	63/63	52/52	45/50	54/54	62/62	56/56
The average age of cases/control (years) at the diagnosis of DM	5.9/6	5.3/5	6.5/6.5	8.1/7.4	4/NA	7.97/7.91
The average age of cases (years) at the diagnosis of CD	9.1	10.8	10	13.8	7.5	11.3
Follow up after diagnosis of CD (years)	2	2	1	4	1	1
The quality assessment GRADE system	Moderate	Moderate	Moderate	Low	Low	Low
Number of patients baseline/follow up	98/98	67/67	293/269	33/33	63/63	48/48

T1DM: type 1 diabetes mellitus; CD: celiac disease.

**Table 2 children-09-01247-t002:** Effects of GFD in pediatric patients with a dual diagnosis of T1DM and asymptomatic CD.

Study	[N]	Effect of GFD	*p*-Value
Warnecke et al., 2016 [16] *	974	Improvement of HDL levels Improvement in HA1C Improvement in BMI SDS.	< 0.01 <0.0001 <0.0001
Nagl et al., 2019 [17]	608	Improvement in Height SDS Improvement in Weight SDS	0.001 0.001
Salardi et al., 2017 [18]	201	Improvement in Total Cholesterol Improvement in Triglyceride	<0.025 <0.005
Hansen et al., 2006 [19]	33	Improvement in Weight SDS Mean Corpuscle Volume Hemoglobin	0.002 0.02 0.001
Gutch et al., 2016 [20]	24	Improvement in Weight SDS Improvement in Hemoglobin Improvement in serum Iron Improvement in Calcium	<0.05 <0.05 <0.05 <0.05
Acerini et al., 1998 [21] *	7	Effect on growth parameters or glycemic control	NS
Taler et al., 2012 [27]	68	No significant effect on growth and glycemic control	NS
Goh et al., 2010 [28]	29	There was no significant effect on growth parameters or glycemic control.	NS
Westman et al., 1999 [29]	20	Effect on growth parameters or glycemic control	NS
Sanchez-Albisua et al., 2005 [30]	9	Improvement in Height SDS.	0.03
Sponzilli et al., 2010 [31] *	12	Increase in Insulin requirement Increase in HA1C	0.02 <0.001
Valerio et al., 2008 [32]	57	Bone density improved significantly with strict glycemic control and adherence to GFD.	0.015
Malalasekera et al., 2009 [33]	21	Shows renoprotection of GFD	0.04
Gopee et al., 2013 [34]	24	Shows renoprotection of GFD	0.01
Mohn et al., 2001 [35]	20	Increase in Insulin requirements	0.05
Frohlich-Reiterer et al., 2011 [36]	411	Improvement in Weight SDS Improvement in Height SDS.	0.001 0.001
Narula et al., 2009 [37] *	8	Improvement in Weight-SDS Improvement in BMI-SDS	0.008 0.02
Abid et al., 2011 [38]	22	Increase in Insulin requirements	<0.005
Pham-short et al., 2014 [39]	129	Non-adherence to GFD was associated with early evidence of renal disease	0.04

N = number of patients with the dual diagnosis of T1DM and CD in the study. NS: Not significant. * Differences detected in statistics used; excluded from the forest plot.

## Data Availability

Data is available in Appendix A.

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
