# Peer review of "Does a Gluten-Free Diet Affect BMI and Glycosylated Hemoglobin in Children and Adolescents with Type 1 Diabetes and Asymptomatic Celiac Disease? A Meta-Analysis and Systematic Review"

_children, 2022, doi:10.3390/children9081247_

Round 1

Reviewer 1 Report

Reviewed meta-analysis and systematic review conducted by Burayzat S et al. aimed to assess the effectiveness of long-term gluten-free diet (GFD) in patients with T1DM and CD in preventing complications associated with these chronic diseases.

Below are my comments:

Introduction

1.      The aim of the study, described by the authors as “... proves that they assumed the hypothesis that the GFD diet may positively influence glucose parameters and metabolic factors in patients with T1DM and CD. In my opinion, Introduction does not sufficiently explain the hypothesis of the mata-analysis. Please, could the Authors provide more information in this regard? In clinical practice, it is often observed that the GFD followed by patients is not properly balanced or even has a higher glycemic index than a diet containing gluten. Could the Authors explain why the GFD could have a positive impact in this case? 2.      Page 3, Line It is unclear whether the institution of GFD in T1DM and subclinical CD patients is beneficial. 3.      Please, introduce the abbreviation GFD the first time you use it in the text (Page 2, line 41) and use it consistently throughout the entire manuscript (e.g. page 2 line 50). 4.      There is lots of missing spaces in entire manuscript  e.g. page 2, Line 42; page 3 line 56; page 6 line 129. Please check and correct it. 5.      In page 3, Line 57-58 the Authors desrcibed: „Despite many screening studies on  this subject, only a few have tackled the effects of putting T1DM and subclinical CD on a GFD.”, but there is no citations. Please add the appropriate references. 6.      The same situation in next line (page 3, line 59): „Only a handful of small prospective and retrospective studies have addressed the glycemic benefits of a GFD.” and no references. 7.      Page 4, line 83, 84 – words marked yellow. The same: page 7 line 135-136 and 152-154 and in entire manuscript.  

Materials and Methods

Materials and Methods and Statistical Analysis is sufficiently described. 8.      In page 4, line 78 the Authors noticed: „The inclusion criteria were as follows: children's ages were between one and eighteen years”. But in page 5, line 100 they wrote: „The only rules applied were studies on Humans, language (English and French studies included), and age from zero to eighteen years.” – Could the Authors explain this differences?   Results. 9.      Page 7, line 138 – in Table 1? 10.  Page 7, line 138: “Rami et al. (13) study had the highest number of participants (293 at baseline and 269 at follow-up), while Amin et al. (14) study had the least number of participants (33)”. Maybe it will be better to change on: „Rami et al. (13) study had the highest number of participants (n-293 at baseline and n-269 at follow-up), while Amin et al. (14) study had the least number of participants (n-33)” because it is difficult to distinguish between what is the number of participants and what is the reference number. 11.  Page 7, line 145 – “et al” is in italics 12.  Page 8, section 13.  The Authors wrote: “As a secondary outcome, improving lipid profile after strict adherence to GFD for at least six months was seen in three studies(17–19). Hemoglobin and serum iron improved significantly after being on GFD for at least six months.(20,21) table (2).” 14.  Could provide more detailed information on the effects on lipids e.g. which lipids (total cholesterol, HDL-C, LDL-C, TG?) was improved by GFD? 15.  Typos in second sentence: “months.(20,21) table (2)” – in Table 2?   Discusion Overal, Discusion is well written and property constructed. 16.  In Discusion, page p, line 177: “In this systematic review, GFD did not significantly affect BMI-SDS and HA1c yet showed the benefits of this specific diet on other health aspects such as lipid profile, diabetic retinopathy, and nephropathy.” But in Results there is no section about retinopathy and nephropathy.   Conclusion Conclusion are quite simple but correctly reflect the obtained results of the analysis.

Author Response

Reviewer 1

  1. The aim of the study, described by the authors as “... proves that they assumed the hypothesis that the GFD diet may positively influence glucose parameters and metabolic factors in patients with T1DM and CD. In my opinion, Introduction does not sufficiently explain the hypothesis of the mata-analysis. Please, could the Authors provide more information in this regard? In clinical practice, it is often observed that the GFD followed by patients is not properly balanced or even has a higher glycemic index than a diet containing gluten. Could the Authors explain why the GFD could have a positive impact in this case?
  2. Page 3, Line It is unclear whether the institution of GFD in T1DM and subclinical CD patients is beneficial.

Response for points 1&2:

As the children in this study have a dual diagnosis of celiac disease and diabetes mellitus, specialized dietary regimens are a cornerstone in the management of their conditions. Both CD and T1DM have effects on growth and glycemic control. GFD proved to be of benefit to the growth and nutrient absorption in patients with CD. In cases of CD patients that have been diagnosed with T1DM, will GFD have the same effect on growth and nutrients absorption

The aim of the study “The primary objectives were to document the pooled effects of GFD on anthropometric parameters, (positive effects or negative) mainly body mass index (BMI) and glycosylated hemoglobin (HbA1c) levels for this group of patients”.

The objective was to assess the effect of GFD to help physicians in managing these patients with the dilemma of putting children with T1DM and subclinical CD on GFD, taking into consideration the high glycemic index that this diet might have if it was unbalanced.

Lines 85-86/92-94

  1. Please, introduce the abbreviation GFD the first time you use it in the text (Page 2, line 41) and use it consistently throughout the entire manuscript (e.g. page 2 line 50).

Corrected in lines 4/50/76

  1. There is lots of missing spaces in entire manuscript g. page 2, Line 42; page 3 line 56; page 6 line 129. Please check and correct it.

The entire manuscript was reviewed, and extra spaces removed

  1. In page 3, Line 57-58 the Authors desrcibed: „Despite many screening studies on this subject, only a few have tackled the effects of putting T1DM and subclinical CD on a GFD.”, but there is no citations. Please add the appropriate references.
  2. The same situation in next line (page 3, line 59): „Only a handful of small prospective and retrospective studies have addressed the glycemic benefits of a GFD.” and no references.

Response to comments 5&6. Reference number 7 was added line 97

  1. Page 4, line 83, 84 – words marked yellow. The same: page 7 line 135-136 and 152-154 and in entire manuscript.

Yellow removed

  1. In page 4, line 78 the Authors noticed: „The inclusion criteria were as follows: children's ages were between one and eighteen years”. But in page 5, line 100 they wrote: „The only rules applied were studies on Humans, language (English and French studies included), and age from zero to eighteen years.” – Could the Authors explain this differences?

This was corrected. Ages of patients in the study were from 1-18. Line 144 

  1. Page 7, line 138: “Rami et al. (13) study had the highest number of participants (293 at baseline and 269 at follow-up), while Amin et al. (14) study had the least number of participants (33)”. Maybe it will be better to change on: „Rami et al. (13) study had the highest number of participants (n-293 at baseline and n-269 at follow-up), while Amin et al. (14) study had the least number of participants (n-33)” because it is difficult to distinguish between what is the number of participants and what is the reference number.

Done. Lines 188-189

  1. Page 7, line 145 – “et al” is in italics

Corrected. line 190

  1. Page 8, section 13. The Authors wrote: “As a secondary outcome, improving lipid profile after strict adherence to GFD for at least six months was seen in three studies(17–19). Hemoglobin and serum iron improved significantly after being on GFD for at least six months.(20,21) table (2).” 14.  Could provide more detailed information on the effects on lipids e.g. which lipids (total cholesterol, HDL-C, LDL-C, TG?) was improved by GFD?

Lines 228. Lipid profile elements affected by GFD in these studies were mentioned

  1. Typos in second sentence: “months.(20,21) table (2)” – in Table 2?      Corrected
  2. In Discusion, page p, line 177: “In this systematic review, GFD did not significantly affect BMI-SDS and HA1c yet showed the benefits of this specific diet on other health aspects such as lipid profile, diabetic retinopathy, and nephropathy.” But in Results, there is no section about retinopathy and nephropathy.

The section on retinopathy and renopathy was added to the results.

Conclusions are quite simple but correctly reflect the obtained results of the analysis.

Reviewer 2 Report

Burayrat et al performed a meta-analysis on the interest of gluten free diet in patients with both coeliac disease and type 1 diabetes in modulating BMI and HA1C hemoglobin.

The study was performed according to the standards in such case. Biases were taken into account as far as possible. All the articles kept for analysis were published after 2000. It is of interest that no the authors shed light on the lack of  consensus on the question they analyzed.

Author Response

Reviewer 2:

The study was performed according to the standards in such case. Biases were taken into account as far as possible. All the articles kept for analysis were published after 2000. It is of interest that no the authors shed light on the lack of consensus on the question they analyzed.

This was elaborated on. Line 73

Round 2

Reviewer 1 Report

I accept the manuscript in the corrected version.

This manuscript is a resubmission of an earlier submission. The following is a list of the peer review reports and author responses from that submission.

Round 1

Reviewer 1 Report

This manuscript describes a meta-analysis of studies looking at gluten free diets in children with T1DM and coeliac disease.  Given the common auto-immune basis and co-occurrence of these conditions the topic is relevant but as noted the information is limited which limits the ability to perform meta-analysis.

I do not understand how the selected studies were described as cohort studies on line 174 and case-control studies in line 176.  Which were they?

If children have been diagnosed with CD, whether symptomatic or not, surely it is mandatory to recommend a GFD?  So is this study comparing these children with those having T1DM and not CD to see whether the GFD avoids any detrimental effects of CD?  Don’t you need to have a group with T1DM and CD not following a GFD to assess whether the GFD is useful?   But this would be unethical.

Something wrong in Table 1, the data in rows does not match row description.  Also the study by Saadah et al is in Australia not Canada.

Line 197-8 ‘The patients' age at the diagnosis of CD varied among the studies from 6.5 years in Rami et al.(13) to 11.1 years in Sun et al.(11).’ But line 207-8 ‘while the range of diagnosis age for CD was between six and eight years.’  These two statements are not consistent and do not agree with the Table as I interpret it.

Line 230-232 ‘the overall effect of six cohort studies indicated that there was a statistically significant  difference in the mean [mean -0.08 (95% -0.23, 0.07)] of SDS BMI of T1DM patients between  baseline and the follow-up (P=0.72).’ How do these figures reflect a statistically significant difference?

Check refs 13, 15, 48.

Author Response

Dear Editor:

Many thanks to you and to the reviewers for reviewing our manuscript. We have revised the manuscript in light of the useful suggestions and comments from the reviewers. All the corrections have been highlighted in yellow color in the manuscript's document.

We hope our responses to the comments are up to the level of satisfaction.

Yours truly.

The authors

Response to Comments

Reviewer 1

This manuscript describes a meta-analysis of studies looking at gluten-free diets in children with T1DM and coeliac disease.  Given the common auto-immune basis and co-occurrence of these conditions, the topic is relevant but as noted the information is limited which limits the ability to perform meta-analysis.

I do not understand how the selected studies were described as cohort studies on line 174 and case-control studies in line 176.  Which were they?

Response: Corrected, they are all case-control studies. Line 186

If children have been diagnosed with CD, whether symptomatic or not, surely it is mandatory to recommend a GFD?  So is this study comparing these children with those having T1DM and not CD to see whether the GFD avoids any detrimental effects of CD?  Don’t you need to have a group with T1DM and CD not following a GFD to assess whether the GFD is useful?   But this would be unethical.

Response: Thank you for your remark as you have noted it is unethical to have a celiac patient and not put him/her on GFD. Yet the evidence is inconclusive concerning the advantages versus the disadvantages of screening and treating asymptomatic individuals in children already burdened with an established chronic illness. Moreover, It is unclear whether the institution of GFD in T1DM and subclinical CD patients is beneficial. So the general idea of this research is to find a reliable evidence to convince physicians worldwide of GFD benefits in patients with T1DM and asymptomatic CD.

LINES 83-84 and 87

Something wrong in Table 1, the data in rows does not match row description.  Also the study by Saadah et al is in Australia not Canada.

Response: Table (1) re-alighned and Saadah et al country of research was corrected. Table (1)

Line 197-8 ‘The patients' age at the diagnosis of CD varied among the studies from 6.5 years in Rami et al.(13) to 11.1 years in Sun et al.(11).’ But line 207-8 ‘while the range of diagnosis age for CD was between six and eight years.’  These two statements are not consistent and do not agree with the Table as I interpret it.

Response: Due to the malalignment of the table some numbers got mixed up. It was all corrected in table 1 and the text.

Lines: 201-202/211-212/ Table (1).

Line 230-232 ‘the overall effect of six cohort studies indicated that there was a statistically significant  difference in the mean [mean -0.08 (95% -0.23, 0.07)] of SDS BMI of T1DM patients between  baseline and the follow-up (P=0.72).’ How do these figures reflect a statistically significant difference?

Response: In line 228 a typo was found. Corrected. No significant difference. This result is reflected in the discussion line 295

Check refs 13, 15, 48.

Response: References were checked and re-inserted in AMA form.

Reviewer 2 Report

Results are correct and clear, as possibly extected by researchers involved in the field.

But the manuscript is really too large and over-written and should be cut in the different parts at least 30-40%.

There is no need to describe CD again and again either in the introduction as well as in the discussion (lines 252-279)

The methods are excessively detailed and should be condensed. Most is routine.

Results are OK, by I am afraid that Table 1 is not easy to read: it might be splitted.

Par. 4.4 : why not giving an estimate of the benefits of the GFD on the libid, Haemoglobin and Iron profiles % % improvement ??

Discussion: No description of CD again, please.

HA1C reported lower before diagnosis of CD, then improved on GFD ?? Malabsorption is very unlikely to be an explanation before CD diagnosis. Authors should comment about the possible relationship of HA1C with the level of auto-antibodies.

Author Response

Dear Editor:

Many thanks to you and to the reviewers for reviewing our manuscript. We have revised the manuscript in light of the useful suggestions and comments from you. All the corrections have been highlighted in yellow color in the manuscript's document.

We hope our responses to the comments are up to the level of satisfaction.

Yours truly.

The authors

Response to Comments

Reviewer 2

Results are correct and clear, as possibly extected by researchers involved in the field.

But the manuscript is really too large and over-written and should be cut in the different parts at least 30-40%. There is no need to describe CD again and again either in the introduction as well as in the discussion (lines 252-279). The methods are excessively detailed and should be condensed. Most is routine. Discussion: No description of CD again, please.

Response: Redundant information removed

Lines 144-147/151-153/157-161/247-250

Results are OK, by I am afraid that Table 1 is not easy to read: it might be splitted.

Response: Table (1) was reviewed and modified.

Par. 4.4 : why not giving an estimate of the benefits of the GFD on the libid, Haemoglobin and Iron profiles % % improvement ??

Response: When reporting the effect of GFD in table (2) the improvement was significant regardless of percentages. We reported a significant p-Value as evidence of the positive effects of GFD.

Table (2)

HA1C reported lower before diagnosis of CD, then improved on GFD ?? Malabsorption is very unlikely to be an explanation before CD diagnosis. Authors should comment about the possible relationship of HA1C with the level of auto-antibodies.

Response: Malabsorption can be present before a definitive diagnosis of CD; the patient might have manifestations of CD that gradually manifest before a definitive diagnosis.

Fuchs V, Kurppa K, Huhtala H, Mäki M, Kekkonen L, Kaukinen K. Delayed celiac disease diagnosis predisposes to reduced quality of life and incremental use of health care services and medicines: A prospective nationwide study. United European Gastroenterol J. 2018 May;6(4):567-575. doi: 10.1177/2050640617751253. Epub 2018 Jan 8. PMID: 29881612; PMCID: PMC5987279.

HA1C was lower before a diagnosis of CD and it became higher after GFD ( In both situations it was high as these are diabetic patients; it did not improve)

“Only one-third of T1D + CD patients reached constant Ab‐negativity after CD diagnosis. Achieving Ab‐negativity after diagnosis seems to be associated with better metabolic control and growth, supposedly due to higher adherence to therapy in general.”

Nagl K, Bollow E, Liptay S, Rosenbauer J, Koletzko S, Pappa A, Näke A, Fröhlich-Reiterer E, Döring C, Wolf J, Salfeld P, Prinz N. Lower HbA1c in patients with type 1 diabetes and celiac disease who reached celiac-specific antibody-negativity-A multicenter DPV analysis. Pediatr Diabetes. 2019 Dec;20(8):1100-1109. doi: 10.1111/pedi.12908. Epub 2019 Aug 26. PMID: 31430021; PMCID: PMC6899993.

Round 2

Reviewer 1 Report

Some improvements have been made to the manuscript.

Overall the manuscript could be improved by editing by a native English speaker. 

In discussion there are multiple paragraphs with one ( eg 253-255) or two (eg line 248-252) sentences.  The whole discussion needs to be rewritten to make proper paragraphs.

Is ‘prometric’ an appropriate word for here?  I cannot find any definition that makes sense here.  Do you mean anthropometic?

Lines 205, 217 still refers to cohort studies.

Line 207, the ‘overall pooled…’ what should be go here? Same with line 212, 219.

Are the values in the results section on HbA1C ‘mean differences’?  The numbers have been labelled ‘mean’ but I suspect this is not correct.

Lines 238-40, ‘In patients with a dual diagnosis of CD and T1DM, the classic form of CD may be observed in less than 25 % of T1DM patients, with typical gastrointestinal symptoms such as chronic diarrhea, body mass  deficiency, abdominal pain, and flatulence.’ Reorder to ‘In patients with a dual diagnosis of CD and T1DM, the classic form of CD with typical gastrointestinal symptoms such as chronic diarrhea, body mass  deficiency, abdominal pain, and flatulence, may be observed in less than  25 % of T1DM patients.’

Line 274, 279, 289 HA1C should be HbA1c.

First para of discussion should summarise your findings.

Ref 15 is wrong.  There is no author Article O on this paper as far as I can see.

Ref 48 looks incorrect.

Table 1 has been corrected to show that the study by Saadah et al was in Australia but this was not corrected in the text line 183.

The study by Mackinder is noted to be in England but the centre was a hospital in Glasgow which is in Scotland.  Could use UK but that is not same as England.

Author Response

Response to the reviewers’ comments  2

Thank you for your time and effort in reviewing our manuscript. The comments were of high precision and value.

Overall the manuscript could be improved by editing by a native English speaker.

Grammarly software was used for English editing, with a report of 99% correct. Report can be provided.

In discussion there are multiple paragraphs with one ( eg 253-255) or two (eg line 248-252) sentences.  The whole discussion needs to be rewritten to make proper paragraphs.

The discussion was re-written in a more coherent way. And small paragraphs with close ideas combined.

Lines 250-253/262-263/272-273/285-288/327-330.

Is ‘prometric’ an appropriate word for here?  I cannot find any definition that makes sense here.  Do you mean anthropometic?

Prometric changed to anthropometric in three positions. Lines 121-122/262

Lines 205, 217 still refers to cohort studies.

The cohort was omitted. Lines 205/217/335/538

Line 207, the ‘overall pooled…’ what should be go here? Same with line 212, 219.

This was corrected. Lines 207/212/219/223

Are the values in the results section on HbA1C ‘mean differences’?  The numbers have been labelled ‘mean’ but I suspect this is not correct.

Mean difference was corrected . Lines 218/222

Lines 238-40, ‘In patients with a dual diagnosis of CD and T1DM, the classic form of CD may be observed in less than 25 % of T1DM patients, with typical gastrointestinal symptoms such as chronic diarrhea, body mass  deficiency, abdominal pain, and flatulence.’ Reorder to ‘In patients with a dual diagnosis of CD and T1DM, the classic form of CD with typical gastrointestinal symptoms such as chronic diarrhea, body mass  deficiency, abdominal pain, and flatulence, may be observed in less than  25 % of T1DM patients.’

Done. Lines 238-40

Line 274, 279, 289 HA1C should be HbA1c.

Done. Lines 52/270/271/289/538

First para of discussion should summarise your findings.

The first paragraph in the discussion re-written. Lines 232-8

Ref 15 is wrong.  There is no author Article O on this paper as far as I can see.

Corrected. Reference 15

Ref 48 looks incorrect.

Reference 48 corrected

Table 1 has been corrected to show that the study by Saadah et al was in Australia but this was not corrected in the text line 183.

Corrected. Line 183

The study by Mackinder is noted to be in England but the centre was a hospital in Glasgow which is in Scotland.  Could use UK but that is not same as England.

Corrected. Lines 182-3
